# Prognostic Significance of Computed Tomography Findings in Pulmonary Vein Stenosis

**DOI:** 10.3390/children8050402

**Published:** 2021-05-17

**Authors:** Laureen Sena, Ryan Callahan, Lynn A. Sleeper, Rebecca S. Beroukhim

**Affiliations:** 1UMass Memorial Children’s Medical Center, Department of Radiology, University of Massachusetts Medical School, Worcester, MA 01655, USA; laureen.sena@umassmemorial.org; 2Department of Cardiology, Boston Children’s Hospital, Harvard Medical School, Boston, MA 02115, USA; ryan.callahan@cardio.chboston.org (R.C.); lynn.sleeper@cardio.chboston.org (L.A.S.)

**Keywords:** pulmonary vein stenosis, pulmonary vein atresia, pulmonary vein compression, computed tomography, pulmonary venous collaterals, perihilar induration, pulmonary cysts

## Abstract

(1) Pulmonary vein stenosis (PVS) can be a severe, progressive disease with lung involvement. We aimed to characterize findings by computed tomography (CT) and identify factors associated with death; (2) Veins and lung segments were classified into five locations: right upper, middle, and lower; and left upper and lower. Severity of vein stenosis (0–4 = no disease–atresia) and lung segments (0–3 = unaffected–severe) were scored. A PVS severity score (sum of all veins + 2 if bilateral disease; maximum = 22) and a total lung severity score (sum of all lung segments; maximum = 15) were reported; (3) Of 43 CT examinations (median age 21 months), 63% had bilateral disease. There was 30% mortality by 4 years after CT. Individual-vein PVS severity was associated with its corresponding lung segment severity (*p* < 0.001). By univariate analysis, PVS severity score >11, lung cysts, and total lung severity score >6 had higher hazard of death; and perihilar induration had lower hazard of death; (4) Multiple CT-derived variables of PVS severity and lung disease have prognostic significance. PVS severity correlates with lung disease severity.

## 1. Introduction

Pulmonary vein stenosis (PVS) is a rare but serious disease in infants and children that can be isolated, or associated with congenital heart disease or chronic lung disease of prematurity. The incidence of PVS in children is reported to be in the range of 0.0017–0.03%, with a bimodal distribution of age at diagnosis, with 18 months as a dividing point [1,2]. The condition is often progressive, leading to pulmonary hypertension, respiratory failure, and death [3]. Although PVS has been well described in high-income countries, recent data have shown that PVS also affects children in low- to middle-income countries, with a high mortality [4]. In spite of advances in medical and surgical management of the disease, patients are at risk for restenosis and may require repeated interventional catheterizations and surgeries [5,6]. Although outcomes are excellent for single-vessel pulmonary vein stenosis, mortality can be significantly higher when there is multivessel involvement [7,8]. Because of the high acuity of this disease, noninvasive techniques for assessment of clinical status and disease progression are needed to guide treatment and prognosis. Although assessment of PVS by computed tomography (CT) has been shown to be a reliable technique for evaluation of PVS in children, prior research has focused on evaluation of the anatomy and structure of the individual pulmonary veins [9,10,11]. However, the pathology of lung disease associated with PVS as assessed by CT is not well reported. An improved understanding of the role of CT findings could help clinicians in predicting patient outcome and guiding management [12]. Therefore, we aimed to (1) describe pulmonary vein and lung findings by contrast-enhanced CT and (2) identify CT factors associated with death.

## 2. Materials & Methods

Subjects: A database search at Boston Children’s Hospital identified patients who were evaluated by CT for a diagnosis of pulmonary vein stenosis between February 2007 and August 2020. Studies with poor delineation of the pulmonary veins or excessive hardware (e.g., indwelling stents) that obscured evaluation of pulmonary vein or lung anatomy were excluded. This retrospective study was approved by the Boston Children’s Hospital Institutional Review Board with waiver of consent. The data that support the findings of this study are available from the corresponding author upon reasonable request.

Clinical data: Clinical records were reviewed for demographic data and past medical history. Medical history elements included history of prematurity, aspiration, gastroesophageal reflux disease, and genetic diagnoses. Cardiac history included prior history of congenital heart disease.

Imaging: CT images were reviewed for pulmonary vein characteristics, including presence and severity of stenosis, and specific anatomy that could lead to pulmonary vein stenosis, such as compression from the aorta, bronchus, or pulmonary artery. Evidence of pleural thickening and interlobar pulmonary vein collateral formation were recorded. Mediastinal induration (low-density amorphous soft tissue surrounding mediastinal structures), mediastinal lymphadenopathy, and perihilar induration (low-density amorphous soft tissue surrounding hilar structures) were also recorded. Lung characteristics included interlobular septal thickening, ground glass opacity, and cyst formation. An individual lobe or entire lung was categorized as hypoplastic when the lobe or lung, including the supporting pulmonary artery and peripheral branches, was smaller than the other lobes or the contralateral lung, respectively.

PVS severity score: We used a modified PVS severity score, based on prior work from Kalfa et al. [7]. A PVS severity was assigned to each vein using CT findings only (right upper, right middle, right lower, left upper, and left lower): 0 = no disease; 1 = mild focal disease; 1.5 = mild diffuse disease; 2 = moderate disease; 2.5 = moderate diffuse disease, 3 = severe focal disease; 3.5 = severe diffuse disease, and 4 = atresia (lack of continuity of a vein and diffuse hypoplasia or small caliber vein in the periphery). Mild disease was defined as <50% segmental narrowing, moderate disease was 50–90% segmental narrowing, and severe disease was >90% segmental narrowing. Focal disease was defined as a discrete narrowing of the vessel near the orifice, or related to external compression; whereas diffuse disease was defined as a diffusely hypoplastic vessel. If a common left lingular/upper or common left lingular/lower lobe pulmonary vein was present, the vein was labeled an upper or lower vein. The PVS severity score was calculated as the sum of each pulmonary vein score + 2 if bilateral disease was present (maximum score possible = 22; Figure 1).

Lung severity score: A lung severity score was created for the purpose of this study, and was calculated for each lung segment (right upper lobe, right middle lobe, right lower lobe, left upper lobe, left lower lobe, and lingula): 0 = unaffected, no manifestations of pulmonary vein stenosis; 1 = mild, no cyst formation, other manifestations focal or mild; 2 = moderate, no cyst formation, some other manifestations severe or diffuse, ± lung hypoplasia; 3 = severe, any cyst formation, all other manifestations present, diffuse and severe ± lung hypoplasia. A total lung severity score was calculated as the sum of each lung segment score (maximum possible score = 15; Figure 1). Collapsed lung segments were excluded from analysis.

Statistics: Non-normally distributed continuous variables are described as median (IQR), and normally distributed continuous variables are described as mean (standard deviation). Categorical variables are described as frequency (percentage). Univariate Cox proportional hazards regression was used to identify which variables were associated with a higher hazard of death. For Cox regression analysis, age was divided into quartiles. Cutoffs for PVS severity score and total lung severity score were determined based on analysis of ROC curves. A generalized linear model was fit to the data to derive the maximum likelihood estimation of the severity of pulmonary vein stenosis and lung segment score for each of the 5 locations (right upper, right middle, right lower, left upper, and left lower). *p*-values <0.05 were considered statistically significant. Analyses were performed using SAS statistical software (version 9.4, SAS Institute, Inc., Cary, NC, USA).

## 3. Results

Of 94 patients with available CT imaging, we identified 43 first-time CT studies with acceptable image quality, performed between February 2007 and October 2020. Baseline patient characteristics are shown in Table 1. The median age at PVS diagnosis was 11 months, with a median age at CT of 21 months. Of the 14/43 (33%) who were born prematurely, 9/14 (64%) had associated congenital heart disease. Of the 32/43 (74%) with congenital heart disease, the most common diagnosis was totally anomalous pulmonary venous connection (15 patients).

Patient-level pulmonary vein and lung characteristics by CT are shown in Table 2, with references to figures demonstrating features of PVS. Of the 43 patients studied, the median number of veins was 5 (IQR: 4, 5) and the median number of vessels with stenosis was 3.5 (IQR: 2.5, 4). Of the 33 (77%) patients with extrinsic pulmonary vein compression, the most common causes were compression from the descending aorta (left lower pulmonary vein, 18 patients) and compression from a mainstem bronchus (left upper pulmonary vein, 11 patients). Compression leading to stenosis was also associated with dilated main or branch pulmonary arteries, and in several cases, stenosis occurred in pulmonary veins with an oblique orifice relative to the left atrium. Nearly all patients had evidence of interlobular septal thickening and ground glass opacities, with a majority also demonstrating lobar or total lung hypoplasia. Lung cysts were seen in patients with pulmonary vein atresia, and in patients with chronic lung disease of prematurity. Pleural thickening, perihilar and mediastinal induration, and lymphadenopathy were also seen.

Individual pulmonary vein and lung segment characteristics are shown in Table 3. Of the 43 CT examinations, 16 veins were not analyzed because of common right- or left-sided morphology (*n* = 14), or the presence of a Scimitar vein (*n* = 2). Similarly, 13 lung segments could not be analyzed because of lobar collapse. Therefore, 199 individual pulmonary veins and 202 lung segments were assessed for degree of PVS and associated lung findings. There was less severe involvement of the right lower pulmonary vein compared to the other veins (*p* = 0.007), with no significant difference in lung involvement between the six lung segments (*p* = 0.192). When analyzed by individual vein and corresponding lung segment (*n* = 184), there was a correlation between the severity of vessel disease and the severity of lung disease (Figure 2; *p* < 0.001). The median PVS severity score was 9 (IQR 6–12), and the median total lung severity score was 6 (IQR 4–8) (Table 2).

There was 30% mortality by 4 years after the CT. A PVS severity score > 11, presence of cysts, and total lung severity score > 6 were associated with an increased hazard of death, whereas perihilar induration was associated with a decreased hazard of death (Table 4, Figure 3).

## 4. Discussion

Herein, we present data from a detailed analysis of CT findings on a large cohort of children with PVS. In addition to characterizing pulmonary vein disease, we describe several pulmonary abnormalities that can be found in this debilitating disease. Among this analysis of 43 patients, a PVS severity score > 11 and total lung severity score > 6 were both associated with a higher hazard of death in univariate analysis. We also found a strong correlation between PVS severity and associated lung segment severity.

Noninvasive imaging modalities such as CT and MRI have been shown to be both highly sensitive and reliable for the diagnosis of PVS in children, with prognostic capability [10,11,13]. For example, a retrospective analysis of 31 CT and MRI studies demonstrated a strong relationship between number of pulmonary veins involved, and pulmonary vein cross-sectional area with late risk of death after PVS surgery [14]. Other studies have shown that bilateral disease and earlier age at diagnosis predict poor survival in patients with PVS; and that reintervention is associated with improved survival [6,15]. Our observation that a PVS severity score > 11 confers a higher risk of mortality is consistent with this prior data.

## 5. Pulmonary Vein Characteristics

Similar to prior work, we found that many of the stenotic pulmonary veins were compressed by extracardiac structures such as the descending aorta, left mainstem bronchus, or dilated pulmonary artery (Figure 4 and Figure 5); some also had an oblique insertion into the atrium (Figure 6A,B) [16]. We also found that the right lower pulmonary vein tended to be less severely affected and relatively spared of PVS in our study (Figure 5), similar to that seen in prior studies [14,17]. This observation may be attributed to a lower risk of extracardiac compression of the right lower pulmonary vein. In contrast, the left upper pulmonary vein was more commonly stenotic in our study and others. Whereas the normal anatomic relationships of the left superior pulmonary hilum result in a crowded pathway of the left upper pulmonary vein as it courses to the left atrium, passing the left mainstem bronchus and distal main pulmonary artery, the right lower pulmonary vein courses by relatively few structures in normal circumstances. Theoretically, if any of the adjacent structures becomes enlarged (left atrial appendage or main pulmonary artery) or displaced (left main stem bronchus), mass effect with compression leading to stenosis may result. Dilated main and central branch pulmonary arteries associated with baseline pulmonary hypertension may cause a cyclical phenomenon, leading to worsening of pulmonary vein compression and thus worsening pulmonary hypertension. In some cases, an oblique entry of the pulmonary veins into the left atrium resulted in focal stenosis (Figure 6).

The development of pulmonary venous collaterals would be expected to protect against lung damage due to pulmonary hypertension, as they divert pulmonary venous drainage to less obstructed veins. Although we found no significant relationship between the presence of pulmonary venous collaterals and death, our study may have been underpowered to demonstrate such a relationship. By CT, these collateral vessels can be seen as tortuous, abnormally arborizing vessels that cross pleural reflections to an adjacent lobe in the periphery of the lung (Figure 7). Alternatively, they may course around hilar structures more centrally (Figure 8A–C), or both (Figure 5). In addition, we observed mass-like clusters of tiny enhancing vessels between larger interlobar collateral vessels (Figure 5 and Figure 8). These clusters of tiny vessels tend to be located adjacent to pleural reflections, and resemble small areas of consolidation on lung reconstructions. However, the vascular enhancement of these lesions is well visualized on soft tissue reconstructions. These lesions may represent the CT correlate of hemangioma-like foci, which have been found at pathology in patients with PVS [12].

PVS, especially when severe, results in redistribution of pulmonary arterial flow away from the affected lung segment [13]. By CT, we observed differential pulmonary venous contrast enhancement in some patients (increased or decreased enhancement; Figure 4, Figure 5 and Figure 7). This variable pulmonary venous enhancement pattern may be attributed to slow flow in the setting of ostial stenosis (increased enhancement), or from decreased perfusion in severely hypoplastic or atretic veins with flow redistribution away from the lung segment (decreased enhancement). Decreased enhancement may result in difficulty identifying veins on CT, especially with volume-rendered reconstructions (Figure 9A,B). The enhancement of the pulmonary veins is not only dependent on the relative flow distribution to each lung segment, but also depends on the timing of the contrast bolus when the scan is acquired. In addition, there may be preferential enhancement of left- or right-sided pulmonary veins when intravenous contrast is injected into patients with bidirectional Glenn (Figure 10) or Fontan circulations. Therefore, knowledge of the sources of pulmonary artery blood flow and the site of contrast injection should be taken into consideration before attributing variation in pulmonary vein enhancement to PVS. Correlation with radionuclide perfusion lung scans or obtaining iodine maps with dual-energy CT acquisitions at the time of the CTA may help to identify segments with decreased perfusion (Figure 4 and Figure 10) [18]. Timing of the scan acquisition during the intravenous contrast injection can be challenging, especially when patients are tachycardiac and transit times through the circulation are hyperdynamic, particularly in infants. Because of this phenomenon, we suggest a delayed acquisition of at least 25–30 s from the start of contrast injection, to ensure that there is some enhancement of peripherally hypoplastic or atretic veins.

## 6. Lung Characteristics

In prior publications on CT imaging findings in PVS, attention has traditionally been focused on characterization of the degree of stenosis of the individual pulmonary veins. However, in this manuscript, we also describe a constellation of lung findings associated with pulmonary venous obstruction. It is important to note that many of these lung findings could be considered nonspecific or inaccurately ascribed to other disorders when evaluated in isolation. For example, many of our patients had comorbidities such as chronic lung disease of prematurity, gastroesophageal reflux disease, and aspiration that may cause similar pulmonary findings. Nonetheless, we found a high prevalence of lung abnormalities in our patient cohort irrespective of their associated disorders (Table 2). Moreover, we found a significant correlation between the severity of individual-vein PVS and the respective lung lobe severity (Figure 2). This correlation suggests that the impact of PVS on the lung parenchyma adds to baseline lung disease attributable to underlying conditions. Therefore, knowledge of the clinical condition of the patient and their comorbidities at the time of the scan is critical. Furthermore, correlation with available perfusion lung scans and overall awareness of which pulmonary veins are stenotic should improve diagnostic capability [18].

We identified a high lung severity score as a univariate predictor of death. Children with PVS have multiple findings that contribute to the lung severity score, including interlobular septal thickening, ground glass opacity, lung segment and pulmonary artery hypoplasia, pleural thickening, mediastinal and perihilar induration, and cysts. Interlobular septal thickening and ground glass opacity were seen in the majority of patients, and are consistent with interstitial pulmonary edema from chronic pulmonary venous hypertension. Diffuse pleural thickening tends to be seen around lobes or entire lungs involved with more severe PVS (Figure 11). Lobar or total lung abnormalities associated with severe PVS with atresia included diffusely small size of the affected lobe or lung together with decreased size of the feeding pulmonary artery and decreased vascularization (Figure 12). Segmental lung or pulmonary artery hypoplasia could be attributed to decreased growth of the lung due to chronic hypoperfusion.

Notably, we found that perihilar induration (Figure 8) was associated with a lower hazard of death. We hypothesize that the perihilar induration, mediastinal induration, and diffuse pleural thickening associated with PVS on CT could reflect manifestations of secondary lymphatic proliferation and increased lymphatic drainage of edematous lung, which may have a relative protective effect. The lymphatic drainage of the lung is primarily along peribronchovascular bundles and pleural lymphatics, which drain to the central lymphatics of the mediastinum. The pulmonary lymphatic system provides clearance of interstitial fluid and is a necessary component of postnatal pulmonary function. Furthermore, lymphatic development is thought to continue postnatally through sprouting of new vessels from existing lymphatic vessels or circulating progenitor lymphatic endothelial cells [19]. Lymphatic abnormalities such as pulmonary lymphangiectasia have been documented in 50% of lung biopsies in patients undergoing surgical repair of PVS, suggesting that abnormalities involving the lymphatic system are widespread in this disease [12].

Central intraparenchymal and peripheral lung cysts were found in patients with severe PVS with atresia (Figure 10 and Figure 12), reflecting severe lung damage and fibrosis. Similar to other pulmonary findings in PVS, cysts are also nonspecific and can be seen in other forms of pulmonary disease such as chronic lung disease (Figure 8 and Figure 9), pulmonary interstitial glycogenosis, primary surfactant deficiency disorders in term infants, and certain genetic conditions such as Trisomy 21 [20,21]. Therefore, the distribution of lung cysts and knowledge of any underlying confounding conditions must be considered before lung cysts can be attributed to lung damage associated with PVS.

Incidentally, several of the CT studies also exhibited scattered hyperlucent lobules throughout the lung parenchyma, which we believe is related to prematurity rather than PVS (Figure 6E,F and Figure 8). Chronic lung disease of prematurity, or bronchopulmonary dysplasia, is an alveolar growth disorder characterized by hyperlucent lobules of the lung that have decreased vascularization, which results from alveolar enlargement and simplification as well as interstitial fibrosis with interlobular septal thickening [20]. The known association between prematurity and development of PVS increases the likelihood that some patients had both chronic lung disease of prematurity, and other classic pulmonary findings of PVS [17]. Hyperlucent lobules, therefore, should not be confused with other more characteristic pulmonary findings in PVS described above.

## 7. Study Limitations

Although we identified multiple factors associated with death in univariate analysis, our small sample size limited our power to perform multivariable analyses. Similarly, our sample size did not permit us to control for potential confounders of lung disease severity such as prematurity, history of aspiration, and history of gastroesophageal reflux disease. As this was a retrospective cohort study with preselection of CTs based on ease of interpretation, our results may not be generalized to patients with significant hardware. Although scanners and protocols for assessment of PVS were not uniform, as there was a large group of patients referred from other hospitals, this factor may improve overall generalizability. CT scans for PVS are typically performed with a single rapid acquisition and suspension of respiration or free breathing, and are usually not ECG-gated. There is a potential to falsely attribute focal PVS to extrinsic compression from adjacent structures because scans are not dynamic or time-resolved, as could be potentially observed during cardiac catheterization. In addition, the degree of stenosis or atresia of an individual pulmonary vein may be overestimated by CT because of factors that can alter the perfusion through a stenotic vein, which could be overcome with selective or direct contrast injections performed at cardiac catheterization.

## 8. Conclusions

CT is a useful diagnostic modality in the evaluation of patients with PVS. A PVS severity score > 11 and total lung severity score > 6 are associated with death. Clinicians must assess both patient-level factors and available imaging studies to accurately predict severity and risk of adverse outcome. Pulmonary findings in PVS, although nonspecific when taken out of context, strongly correlate with pulmonary vein stenosis severity and should be interpreted with knowledge of the entire clinical picture.

## Figures and Tables

**Figure 1 children-08-00402-f001:**
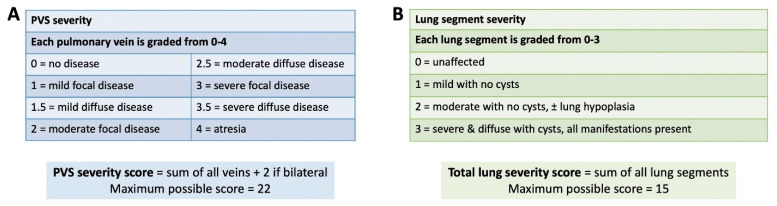
Calculations for pulmonary vein stenosis (PVS) severity score and total lung severity score. PVS severity score was the sum of individual PVS severity scores +2 if bilateral disease was present (**A**), and total lung severity score was the sum of lung segment severity scores (**B**).

**Figure 2 children-08-00402-f002:**
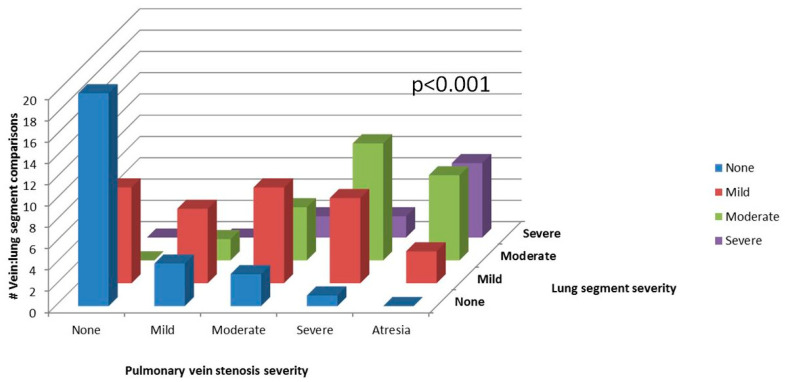
Relationship between pulmonary vein stenosis severity and associated lung segment severity (198 vein and lung segment comparisons).

**Figure 3 children-08-00402-f003:**
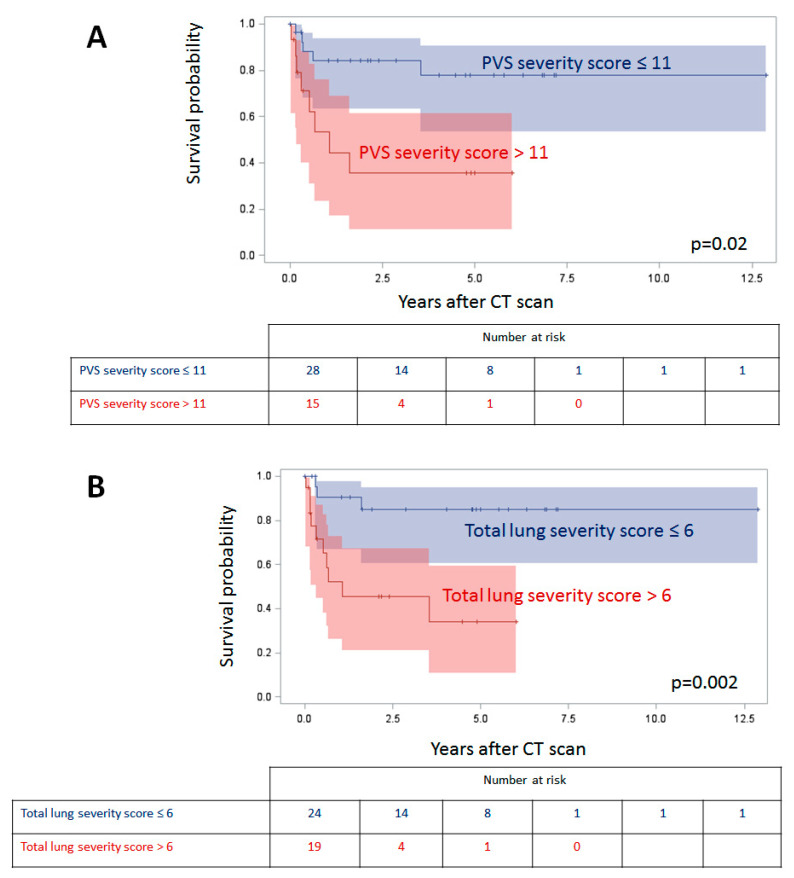
Survival by pulmonary vein stenosis (PVS) severity score (**A**) and total lung severity score (**B**) with 95% confidence intervals.

**Figure 4 children-08-00402-f004:**
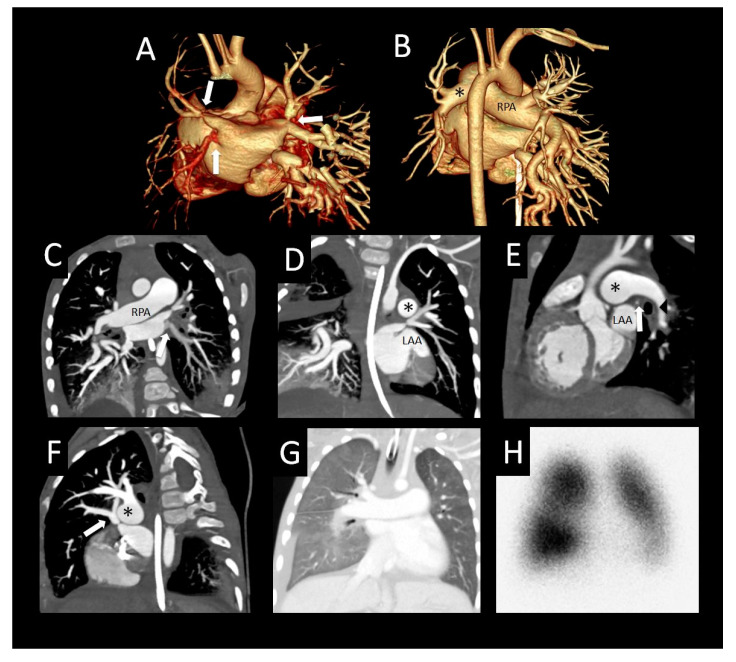
Primary pulmonary vein stenosis. Primary pulmonary vein stenosis diagnosed at 1 year with respiratory distress that was thought to be pneumonia, but failed to improve with antibiotics. Posterior volume rendered CT images show bilateral pulmonary vein stenosis with narrowing of multiple pulmonary veins (arrows), left > right (**A**), dilated right pulmonary artery (RPA), and small distal left pulmonary artery (*) with hypoplastic left hilar branches (**B**). Oblique coronal image (**C**) shows oblique course and decreased enhancement of the left lower pulmonary vein (arrow). Coronal (**D**) and sagittal (**E**) images show severe focal narrowing of the left upper pulmonary vein (arrow) as it courses between the left atrial appendage (LAA), a dilated main/left pulmonary artery confluence (*) and left bronchus (arrowhead). The right upper pulmonary vein also has severe focal narrowing as it courses by the dilated right pulmonary artery (RPA) (*) ((**F**) sagittal oblique image). Coronal lung image (**G**) shows right > left pulmonary edema, a small right pleural effusion, and left lung hypoplasia. Perfusion scintigraphy anterior image (**H**) shows 34% left and 66% right lung perfusion with decreased relative perfusion of the left lower lobe.

**Figure 5 children-08-00402-f005:**
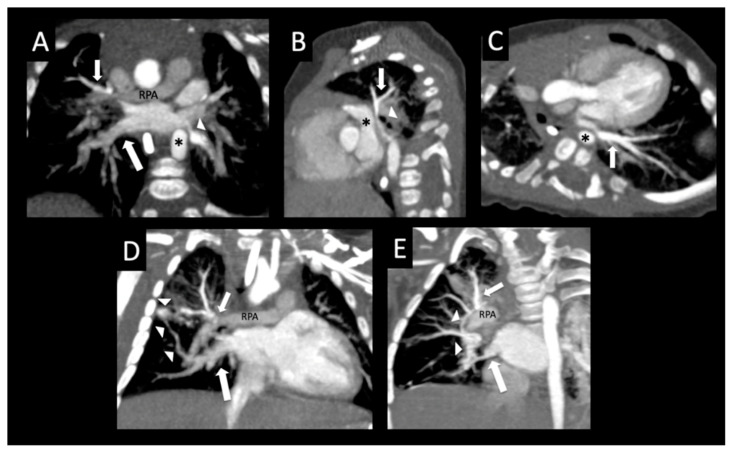
Bilateral pulmonary vein stenosis. Former 35-week triplet with chromosome 2 duplication and primary pulmonary vein stenosis affecting all pulmonary veins except right lower pulmonary vein. Oblique coronal image (**A**) shows moderate ostial stenosis of the left lower pulmonary vein (arrowhead) coursing anterior to the descending aorta (*), atresia of the mid right upper pulmonary vein (small arrow) as it crosses the right pulmonary artery (RPA), and an unobstructed right lower pulmonary vein (large arrow). Sagittal oblique image (**B**) shows extrinsic compression of the left upper pulmonary vein (arrow) coursing between the left atrial appendage (*) and left bronchus (arrowhead). Oblique axial image (**C**) shows moderate extrinsic compression of the left lower pulmonary vein (arrow) from the descending aorta (*). The left upper pulmonary vein (**B**) and left lower pulmonary vein (**A**,**C**) have more dense peripheral enhancement due to central stenosis. Coronal oblique image (**D**) shows focal atresia of the mid right upper pulmonary vein (arrow) at the level of the right pulmonary artery (RPA), and a peripheral network of tiny collaterals (arrowheads) connecting to the right lower pulmonary vein (large arrow). Coronal oblique image (**E**) shows an additional network of intrapulmonary collaterals (arrowheads) coursing around the right pulmonary artery (RPA), which connect the right upper pulmonary vein (small arrow) to the right lower pulmonary vein (large arrow).

**Figure 6 children-08-00402-f006:**
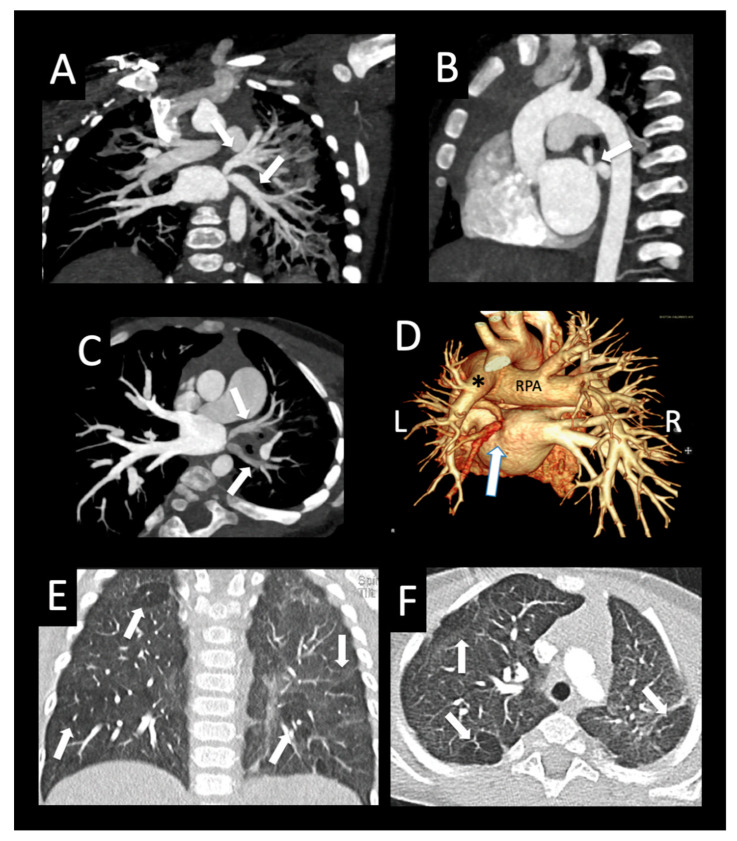
Oblique pulmonary vein course. Former premature infant with left-sided pulmonary vein stenosis and chronic lung disease. Coronal oblique (**A**) and sagittal (**B**) images show oblique, moderately narrowed orifices of the left upper pulmonary vein and left lower pulmonary vein to the superior aspect of the left atrium (arrows). Worsening focal ostial stenosis 4 months later, with decreased enhancement of both left upper pulmonary vein and left lower pulmonary vein (arrows (**C**)) compared to the normal right pulmonary veins. Posterior volume rendered image (**D**) shows a small left pulmonary artery (*) and a large right pulmonary artery (RPA), in addition to the oblique insertion of the left pulmonary veins (arrow). Coronal and axial lung images (**E**,**F**) show bilateral mosaic attenuation of both lungs with hyperlucent lobules (arrows), consistent with alveolar enlargement and simplification associated with chronic lung disease of prematurity.

**Figure 7 children-08-00402-f007:**
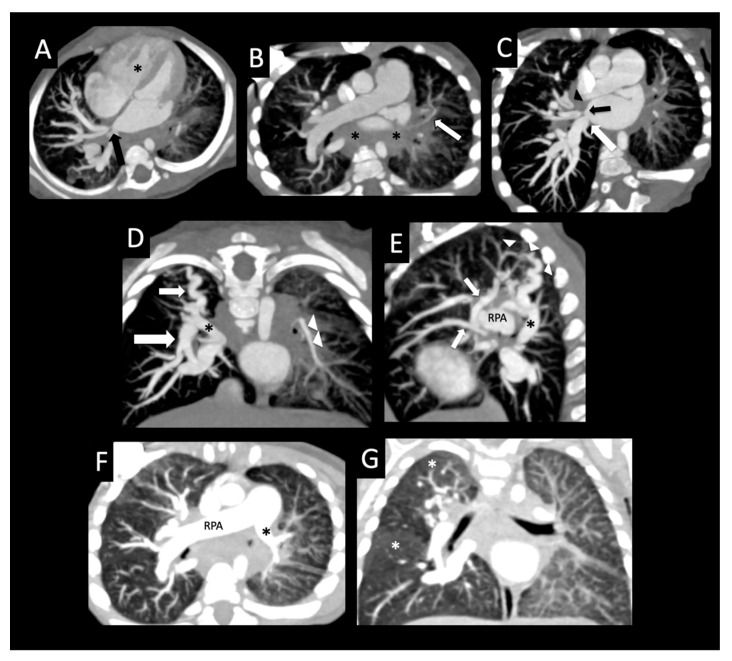
Bilateral pulmonary vein stenosis with collaterals. Congenital dyserythropoiesis with anemia, presenting at 5 months of age with bilateral pulmonary vein stenosis, followed by surgical left common and right upper pulmonary vein repair. CT performed 6 months after surgery shows right ventricular hypertrophy (axial image (**A**)) with septal flattening (*) consistent with pulmonary hypertension. There is severe right middle pulmonary vein ostial stenosis (arrow), and diffusely hypoplastic left lower and left upper pulmonary veins (arrow axial image (**B**)) with bilateral left > right perihilar induration (*). Axial oblique image (**C**) shows ostial right upper pulmonary vein atresia (arrowhead), right middle pulmonary vein severe stenosis (small arrow), and right lower pulmonary vein mild stenosis (large arrow). A large tortuous venous collateral from the right upper lobe (small arrow, (**D**)) connects to the superior segment right lower pulmonary vein (*). The distal right pulmonary artery is large (large arrow) relative to diffusely small distal left pulmonary artery (arrowheads). Sagittal oblique image (**E**) shows peripheral branches of the atretic right upper pulmonary vein (small arrows) with central atresia related to the dilated right pulmonary artery. A collateral network (arrowheads) connects branches of the right upper pulmonary vein to the superior segment right lower pulmonary vein (*), which is unobstructed to the left atrium. Small left lung and left pulmonary artery (*) relative to the right (axial (**F**)). Diffuse interlobular septal thickening and ground glass opacity (consistent with edema) throughout the left lung (coronal image (**G**)), and more mild ground glass opacity involving the right upper and middle lobes (*).

**Figure 8 children-08-00402-f008:**
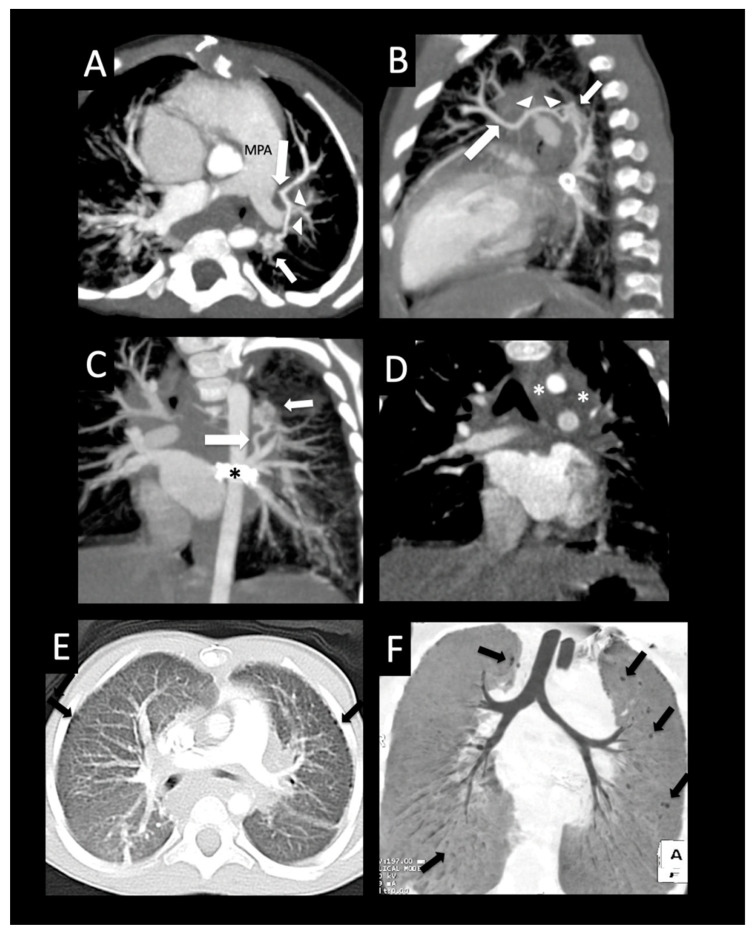
Ex-premature infant with pulmonary vein stenosis. Former 26-week premature infant with chronic lung disease, who underwent sutureless repair of PVS followed by left lower pulmonary vein stenting. Axial (**A**), sagittal (**B**), and coronal (**C**) images from a CTA performed 1 year later show left upper pulmonary vein atresia (large arrow (**A**,**B**)) in proximity to the dilated main pulmonary artery (MPA), with an enhancing interlobar collateral vessel coursing around the left pulmonary artery (arrowheads (**A**,**B**)), which connects to a cluster of vessels (small arrow (**A**–**C**)). Another tortuous collateral (large arrow (**C**)) connects the cluster of vessels to the stented left lower pulmonary vein (***C**). Coronal image (**D**) shows perihilar and mediastinal induration (*). Axial lung image (**E**) shows diffuse interlobular septal thickening, and a coronal minimum-intensity projection (**F**) image shows peripheral and intralobar tiny cysts (arrows (**E**,**F**)). The cysts do not correspond to lung segments most affected by PVS, and may be related to chronic lung disease.

**Figure 9 children-08-00402-f009:**
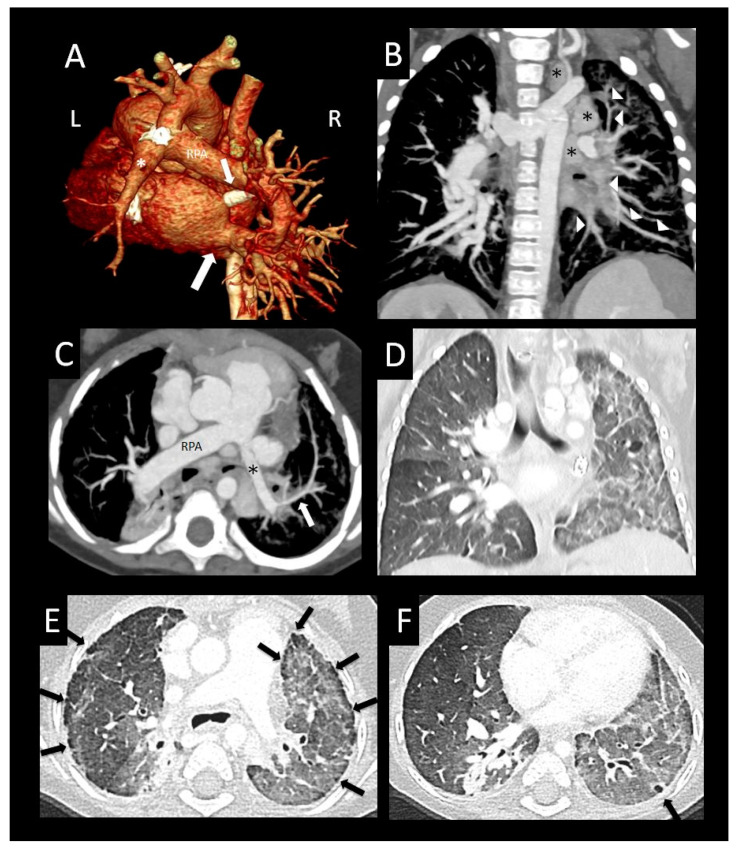
Peripheral cysts in a premature infant with pulmonary vein stenosis. Thirty-four-week premature infant status post atrial septal defect and muscular ventricular septal defect closure, omphalocele repair, and subsequent sutureless repair of the right upper pulmonary vein and left common confluence pulmonary vein. Postoperative CT posterior volume rendered image (**A**) shows no left pulmonary veins entering into the left side of the left atrium consistent with left common pulmonary vein atresia. Note the large right lower pulmonary vein (large arrow) and stented right upper pulmonary vein (small arrow). The peripheral left pulmonary artery (*) and branches are smaller than the right pulmonary artery (RPA). Coronal image (**B**) shows severely hypoplastic peripheral left upper pulmonary vein and left lower pulmonary vein (arrowheads) with diminished enhancement relative to the other vessels with surrounding left-sided perihilar induration. Axial image (**C**) shows large right pulmonary artery and small left pulmonary artery (*) with a hypoplastic branch to the left upper lobe (arrow). Coronal (**D**) and axial (**E**,**F**) images with lung windows show left lung hypoplasia with diffusely increased interlobular septal thickening and ground glass opacity relative to the right lung. The architectural distortion and bilateral peripheral cysts (arrows) involving both lungs are more consistent with chronic lung disease of prematurity than with sequelae of pulmonary vein stenosis.

**Figure 10 children-08-00402-f010:**
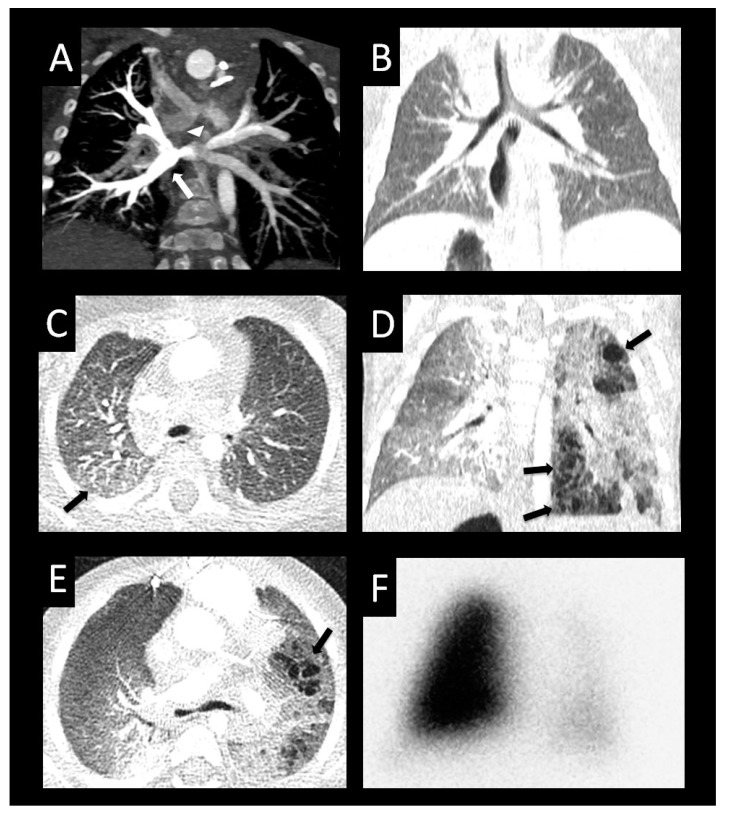
Development of lung cysts in progressive pulmonary vein stenosis. Heterotaxy syndrome, complete atrioventricular canal defect with supracardiac totally anomalous pulmonary venous connection, status post bidirectional Glenn shunt and totally anomalous pulmonary vein repair, with recurrent obstruction of the anastomosis of the central common pulmonary vein confluence to the left atrium. Right arm contrast injection (**A**) shows differential enhancement of the pulmonary veins (right > left), moderate ostial stenosis of the right common pulmonary vein (arrow), and moderate stenosis of the left upper pulmonary vein (arrowhead). Coronal (**B**) and axial (**C**) lung images show normal lungs with mild edema of the right lower lobe (arrow, (**C**)). Coronal (**D**) and axial (**E**) images were performed 5 months after surgical repair of congenital heart disease with right-sided pulmonary vein repair. Note new development of architectural distortion with multiple cysts (arrows) throughout the left lung. The left-sided pulmonary veins were not visualized and were confirmed to be atretic at cardiac catheterization. Anterior perfusion scintigraphy image (**F**) with 11% left and 89% right.

**Figure 11 children-08-00402-f011:**
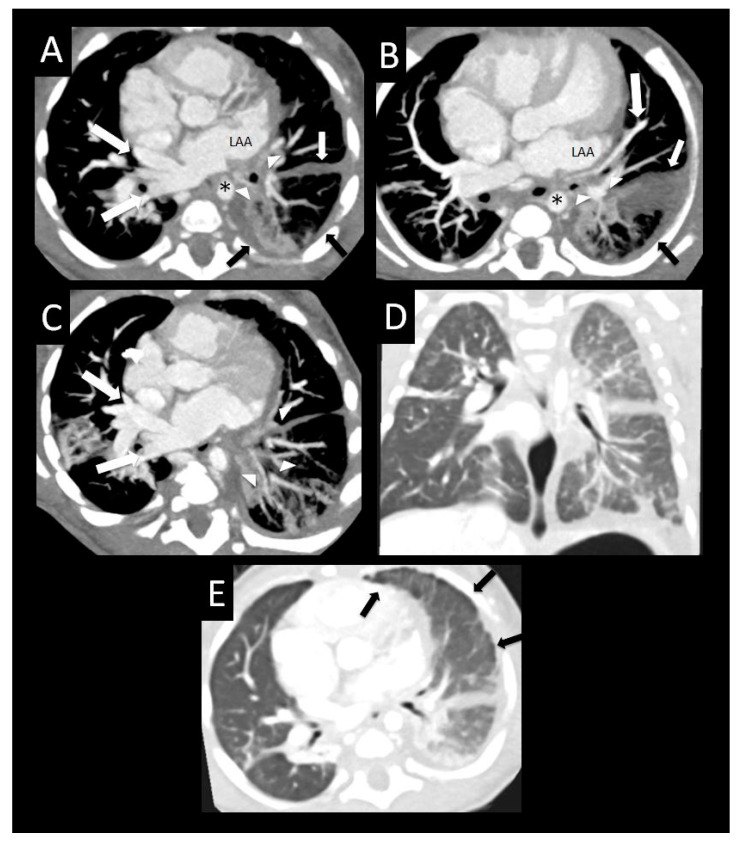
Pulmonary vein stenosis after repair of totally anomalous pulmonary venous connection. Central atretic left pulmonary vein confluence anterior to the descending aorta (*) with peripheral hypoplastic lingular and lower lobe branches (arrowheads) surrounded by perihilar induration on axial imaging (**A**). Diffusely small peripheral left upper pulmonary vein (large arrow) coursing between a severely dilated left atrial appendage and left upper and lower lobe bronchi (arrowheads) on axial imaging (**B**). Left-sided diffuse pleural and fissural thickening (arrows (**A**,**B**)). Diffuse hypoplasia and relative decreased enhancement of the branches of the lingular and left lower pulmonary vein (arrowheads), with large right-sided pulmonary veins on axial imaging (large arrows (**A**,**C**)). Coronal (**D**) and axial (**E**) images with lung windows show diffuse left lung edema with surrounding pleural and fissural thickening (arrows). LAA = left atrial appendage.

**Figure 12 children-08-00402-f012:**
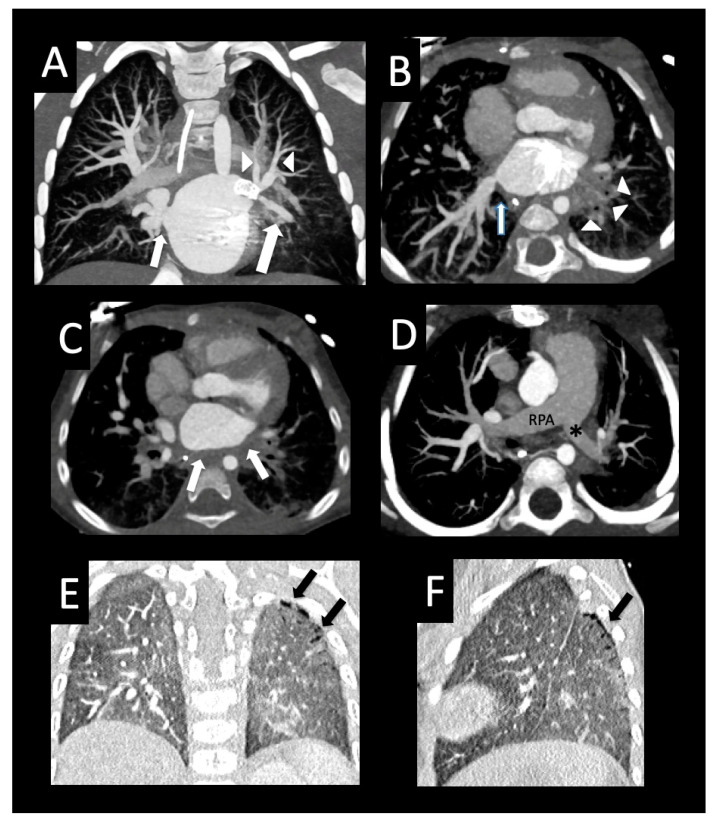
Pulmonary vein stenosis after repair of congenital heart disease. Full-term infant who developed severe bilateral pulmonary vein stenosis after surgical repair of subaortic stenosis, coarctation of the aorta, and ventricular septal defect. The right common pulmonary vein ostium was resected, and the left common confluence was anastomosed to the left atrial appendage with subsequent recurrent left pulmonary vein stenosis and stenting of the left upper pulmonary vein. Follow-up CT shows the left upper pulmonary vein stent (coronal image, (**A**)) extending to the lingular vein (large arrow) that crosses and “jails” the left upper pulmonary vein branches (arrowheads), and mild right upper pulmonary vein ostial stenosis (small arrow, (**A**)). Axial image (**B**) shows moderate stenosis of the right lower pulmonary vein ostium (arrow) and left-sided perihilar induration (arrowheads). The left lower pulmonary vein could not be visualized, consistent with atresia. Axial image (**C**) shows diffuse wall thickening of the posterior left atrium (arrows). The right pulmonary artery (RPA) is dilated relative to the left pulmonary artery (*) (axial image, (**D**)). Coronal (**E**) and sagittal (**F**) images show diffuse decreased vascularity of the left lung relative to the right, and peripheral cysts within the superior segment of the left lower lobe (arrows).

**Table 1 children-08-00402-t001:** Patient characteristics (*N* = 43).

Variable	N (%) or Median (IQR)
Male	24 (56%)
Age at pulmonary vein stenosis diagnosis, years	0.4 (0.2, 0.8)
Age at CT, years	0.7 (0.3, 1.3)
**Race**	
White	21 (49%)
Black	3 (7%)
Asian	1 (2%)
Other	5 (10%)
Unknown	13 (30%)
Structurally normal heart	11 (26%)
Comorbidity (prematurity, congenital heart disease, or genetic diagnosis)	39 (91%)
**Prematurity**	14 (33%)
34–37 weeks gestation	4 (9%)
30–33 weeks gestation	6 (14%)
26–29 weeks gestation	4 (9%)
**Congenital heart disease**	32 (74%)
Totally anomalous pulmonary venous connection	15 (35%)
Supracardiac	6 (14%)
Cardiac	2 (5%)
Infracardiac	3 (7%)
Mixed	4 (10%)
Atrial septal defect	10 (23%)
Patent ductus arteriosus	7 (16%)
Single-ventricle physiology	7 (16%)
Ventricular septal defect	6 (14%)
Atrioventricular canal defect	5 (12%)
Heterotaxy syndrome	5 (12%)
Partially anomalous pulmonary venous connection	4 (9%)
Cor triatriatum	2 (5%)
Hypoplastic left heart syndrome	1 (2%)
Double outlet right ventricle	2 (5%)
Transposition of the great arteries	1 (2%)
Truncus arteriosus	1 (2%)
Other congenital heart disease	16 (37%)
**Genetic diagnosis**	8 (19%)
Trisomy 21	2 (5%)
Other genetic diagnosis	6 (14%)
History of aspiration	13 (30%)
Tracheostomy tube placement	2 (5%)
History of gastroesophageal reflux disease	21 (49%)
Age at death, years (*n* = 13)	1.2 (0.7, 1.6)
Age at follow-up, years	5.7 (2.7, 7.1)

CT = computerized tomography.

**Table 2 children-08-00402-t002:** Patient-level pulmonary vein and lung characteristics by CT (*N* = 43).

Variable	Reference Figures	*N* (%) or Median (IQR) *
**Pulmonary Vein Characteristics**		
PVS severity		Refer to Table 3
No disease	11(A,C)	
Mild disease (<50% segmental narrowing)	7(C)	
Moderate disease (50–90% segmental narrowing)	12(B)	
Severe disease (>90% segmental narrowing)	4(F), 7(C)	
Atresia (lack of continuity and diffuse hypoplasia	7(B)	
Pulmonary vein wall thickening		17 (40%)
Extrinsic compression		33 (77%)
Descending aorta	5(A,C)	18 (42%)
Mainstem bronchus	4(E), 5(B)	11 (26%)
Other	4(F), 5(A,D), 7(E), 8(A,B)	21 (49%)
Intrapulmonary collaterals	5(D,E), 8	20 (47%)
Number of vessels		5 (4, 5)
Number of vessels with stenosis		3.5 (2.5, 4)
Bilateral pulmonary vein involvement		27 (63%)
PVS severity score	1	9 (6, 12)
**Lung characteristics**		
Lung severity		Refer to Table 3
Unaffected		
Mild (no cyst formation, other manifestations focal or mild)	9(G) (right lung)	
Moderate (no cyst, some manifestations severe or diffuse ± lung hypoplasia)	9(G) (left lung)	
Severe (any cyst, all other manifestations present, diffuse and severe ± lung hypoplasia)	10(D), 12(F)	
Interlobular septal thickening	8(E), 9(D,E)	32 (74%)
Ground glass opacity	7(F,G), 9(D,E)	39 (91%)
Lung segment or pulmonary artery hypoplasia	4, 6(D), 7(F), 9(A,C), 12(D)	26 (60%)
Pleural thickening	12	18 (42%)
Mediastinal induration	8(D)	33 (77%)
Mediastinal lymphadenopathy		8 (19%)
Perihilar induration	9(B), 11(A), 12(B)	38 (88%)
Cysts	9, 10, 12(F,G)	10 (23%)
Subpleural cysts		7 (16%)
Intraparenchymal cysts		6 (14%)
Total lung severity score	1	6 (4, 8)

* Data represent number (%) of patients with each finding or median (IQR) among all 43 patients. CT = computerized tomography; PVS = pulmonary vein stenosis.

**Table 3 children-08-00402-t003:** Individual pulmonary vein and lung segment characteristics by CT (N = 199 veins; 202 lung segments). Data expressed as *N* (%) or median (IQR) *.

Pulmonary Veins (*n* = 43 Patients)	All (*n* = 199)	RUPV (*n* = 41)	RMPV (*n* = 33)	RLPV (*n* = 41)	LUPV (*n* = 42)	LLPV (*n* = 42)
Pulmonary vein stenosis severity						
No disease	65 (33%)	11 (27%)	11 (33%)	21 (51%)	11 (26%)	10 (24%)
Mild stenosis (<50% narrowing)	24 (12%)	5 (12%)	4 (12%)	3 (7%)	5 (12%)	7 (17%)
Moderate stenosis (50% narrowing)	37 (29%)	7 (17%)	7 (21%)	8 (20%)	7 (17%)	8 (19%)
Severe stenosis (>50% narrowing)	43 (22%)	11 (27%)	6 (18%)	4 (10%)	12 (29%)	11 (26%)
Atresia (discontinuity)	30 (15%)	7 (17%)	5 (15%)	5 (12%)	7 (17%)	6 (14%)
Focal stenosis	78 (39%)	17 (41%)	14 (42%)	12 (19%)	15 (36%)	20 (48%)
Diffuse stenosis	26 (13%)	6 (15%)	3 (9%)	3 (7%)	9 (21%)	5 (12%)
Pulmonary vein wall thickening	33 (17%)	8 (20%)	6 (18%)	7 (17%)	6 (14%)	6 (14%)
PVS severity score	2 (0, 3)	2 (0, 3.5)	1 (0, 3)	0 (0, 2.5)	2.5 (1, 3.5)	2 (0.5, 3)
**Lung segments (*n* = 41 patients)**	**All (*n* = 202)**	**RUL (*n* = 40)**	**RML (*n* = 39)**	**RLL (*n* = 41)**	**LUL (*n* = 41)**	**LLL (*n* = 41)**
Interlobular septal thickening	94 (47%)	16 (40%)	19 (49%)	15 (37%)	23 (56%)	21 (51%)
Intralobular thickening and ground glass	120 (59%)	24 (60%)	21 (54%)	20 (49%)	28 (68%)	27 (66%)
Lung segment hypoplasia	43 (21%)	3 (8%)	4 (10%)	10 (24%)	13 (32%)	13 (32%)
Lung segment severity score	1 (0, 2)	1 (1, 2)	1 (0, 2)	1 (0, 2)	1 (1, 2)	1 (1, 2)

RUPV = right upper pulmonary vein, RMPV = right middle pulmonary vein, RLPV = right lower pulmonary vein, LUPV = left upper pulmonary vein, LLPV = left lower pulmonary vein, RUL = right upper lobe, RML = right middle lob, RLL = right lower lobe, LUL = left upper lobe, LLL = left lower lobe, PVS = pulmonary vein stenosis. * Data represent number (%) of pulmonary veins or lung segments with each finding, or median (IQR) among all pulmonary veins or lung segments.

**Table 4 children-08-00402-t004:** Univariate Cox regression analysis of variables associated with time to death.

Variable	N	Hazard Ratio	95% CI	*p* Value
Patient characteristics				
Male gender	24 (56%)	0.7	0.2, 2.2	0.56
Age at PVS diagnosis, years				0.63
≤0.18	11 (25%)	1.3	0.2, 8.7	
0.19–0.37	11 (25%)	2.1	0.4, 10.9	
0.38–0.76	11 (25%)	2.7	0.5, 13.4	
>0.76	10 (25%)	Ref		
Age at CT scan, years				0.22
≤0.29	10 (25%)	4.6	0.6, 33.6	
0.30–0.71	11 (25%)	7.1	1, 49.6	
0.72–1.28	11 (25%)	2.8	0.4, 21.9	
>1.28	11 (25%)	Ref		
White race	21 (49%)	0.5	0.1, 2.2	0.35
Prematurity	14 (33%)	1.9	0.6, 5.6	0.25
Congenital heart disease	32 (74%)	0.5	0.2, 1.7	0.27
Single ventricle	7 (16%)	1	0.2, 4.1	0.99
Dextrocardia	4 (9%)	1.9	0.5, 7.8	0.40
Tracheostomy	3 (23%)	1.5	0.3, 7.6	0.60
History of aspiration	13 (30%)	2	0.7, 5.8	0.23
Gastroesophageal reflux disease	24 (56%)	2.5	0.7, 8.7	0.16
Genetic diagnosis	8 (19%)	2.0	0.6, 7.2	0.27
Pulmonary vein findings				
External pulmonary vein compression	33 (77%)	1.1	0.3, 4	0.86
Pulmonary venous collaterals	20 (48%)	0.4	0.1, 1.5	0.18
Number of vessels involved >3	16 (37%)	2.7	0.9, 8.5	0.09
Bilateral pulmonary vein involvement	27 (63%)	4.8	0.8, 28.3	0.08
**PVS severity score > 11**	**15 (35%)**	**4**	**1.3, 12.2**	**0.02**
Lung findings				
Interlobular septal thickening (>3 lung segments)	12 (28%)	2.8	0.9, 8.7	0.07
Intralobular thickening and ground glass (>3 lung segments)	14 (33%)	2.1	0.7, 6.4	0.21
Pulmonary artery and/or lung hypoplasia	25 (58%)	0.4	0.1, 1.3	0.12
**Cyst**	**9 (21%)**	**3.3**	**1.1, 10.1**	**0.04**
Pleural thickening	18 (42%)	0.6	0.2, 2.1	0.44
Mediastinal induration	33 (77%)	0.7	0.2, 2.7	0.62
Mediastinal lymphadenopathy	8 (19%)	1.5	0.4, 5.3	0.56
**Perihilar induration**	**38 (88%)**	**0.3**	**0.1, 1**	**0.04**
**Total lung severity score > 6**	**19 (44%)**	**6.3**	**1.5, 26.8**	**0.01**

## Data Availability

The data presented in this study are available on request from the corresponding author. The data are not publicly available because of limits to the institutional review board agreement at Boston Children’s Hospital.

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
