# Peer review of "Prognostic Significance of Computed Tomography Findings in Pulmonary Vein Stenosis"

_children, 2021, doi:10.3390/children8050402_

Round 1

Reviewer 1 Report

To authors:

The manuscript is well written and easy to follow.

I have minor concerns listed below.

a) The sample size is too small to make a reasonable conclusion that CT findings could provide diagnostic significance in pulmonary valve stenosis.

b) Since CT scan protocol was inconsistent for images, one cannot generalize the results obtained.

c) The patients had several comorbidities such as chronic lung disease of prematurity, gastroesophageal reflux disease and aspiration that may cause similar findings. These CT scans could be separately analysed to actually reflect the effect of PVS.

d) Does PVS severity specifically affects a particular part of lung? Say right or left lobe? or upper or lower lobes of right and left lung?

e) Did the authors observe differences between pulmonary artery and pulmonary vein morphology in CT scans? 

f) Does degree of stenosis correlate with lung severity?

Author Response

Reviewer #1:

Point 1:

The sample size is too small to make a reasonable conclusion that CT findings could provide diagnostic significance in pulmonary valve stenosis.

Response 1:

Thank you for this observation. We agree that our sample size, which includes 43 CT studies and 199 individual pulmonary veins, was relatively small. Despite this challenge, we were able to identify multiple variables associated with time to death by univariate Cox regression analysis because of the relatively high number of events (13 deaths in follow-up; Table 4). However, we were unable to identify variables independently associated with time to death by multivariable analysis due to lack of statistical power. Therefore, our multivariable analysis was omitted from the manuscript. We included the following statement in Study Limitations (page 19, lines 30-31):

‘Although we identified multiple factors associated with death in univariate analysis, our small sample size limited our power to perform multivariable analyses.’

Point 2:

Since CT scan protocol was inconsistent for images, one cannot generalize the results obtained.

Response 2:

Thank you for the comment. We concur that our protocol was inconsistent for image analysis. We chose to omit images with multiple stents and hardware, that would limit our ability to adequately analyze pulmonary vein and lung anatomy. For this reason, we agree that our study lacks generalizability to the entire population of PVS patients. We included the following statement in Study Limitations (page 19, lines 34-38):

‘As this was a retrospective cohort study with pre-selection of CTs based on ease of interpretation, our results may not be generalized to patients with significant hardware. Although scanners and protocols for assessment of PVS were not uniform, as there was a large group of patients referred from other hospitals, this factor may improve overall generalizability.’

Point 3:

The patients had several comorbidities such as chronic lung disease of prematurity, gastroesophageal reflux disease and aspiration that may cause similar findings. These CT scans could be separately analysed to actually reflect the effect of PVS.

Response 3:

Thank you for this observation. A large percentage of our cohort had a history of prematurity (33%), aspiration (30%), and gastroesophageal reflux disease (49%). We agree that these variables may confound the assessment of lung severity in patients with PVS. However, in analysis of the images we recognized several distinguishing features such as the pattern of distribution of lung abnormalities. Unfortunately, our study was underpowered to control for these factors. Therefore, we included the following statement in Lung characteristics (page 16-17, lines 8-10 and 1-6), and in Study Limitations (page 19, lines 31-34):

‘Mosaic attenuation indicative of air trapping (hyperlucent lobules) can be seen in association with chronic lung disease of prematurity, and potentially chronic aspiration associated with gastroesophageal reflux disease (Figure 6). However, the pattern of distribution of lung abnormalities may help to distinguish PVS from these other conditions. For example, lung cysts associated with chronic lung disease of prematurity have a diffuse distribution (Figures 8, 9) while cysts related to severe lung damage from PVS are generally seen in lobs with more severe PVS (Figure 10, 12). In addition, chronic aspiration generally causes infectious or inflammatory opacities that can have a dependent lower lobe distribution.’

‘Similarly, our sample size did not permit us to control for potential confounders of lung disease severity such as prematurity, history of aspiration, and history of gastroesophageal reflux disease.’

Point 4:

Does PVS severity specifically affects a particular part of lung? Say right or left lobe? or upper or lower lobes of right and left lung?

Response 4:

Although we found that PVS severity was lower in the right lower pulmonary vein compared to the other veins, we did not find any significant difference in lung severity between lobes. The following statement was included in the Results (page 5, lines 6-8):

‘There was less severe involvement of the right lower pulmonary vein compared to the other veins (p=0.007), with no significant difference in lung involvement between the 6 lung segments (p=0.192).’

Point 5:

Did the authors observe differences between pulmonary artery and pulmonary vein morphology in CT scans? 

Response 5:

Thank you for the comment. We also felt that the morphology of the pulmonary artery and veins was an important descriptive component of the manuscript. We found lung segment or pulmonary artery hypoplasia in 60% of the studies, and found various characteristics of pulmonary veins, including variations in disease severity as well as focal and diffuse stenosis (Tables 2 & 3).

Point 6:

Does degree of stenosis correlate with lung severity?

Response 6:

This is an important question that we investigated in our study, by using a generalized linear model that was fit to the data to deterive the maximum likelihood estimation of the severity of pulmonary vein stenosis and lung segment score for each of the 5 locations. This statistical technique was used to account for multiple veins analyzed in the same patient. We found a strong correlation between stenosis severity and lung severity (Figure 2). Also, the following statement was included in Results:

‘When analyzed by individual vein: corresponding lung segment (n=184), there was a correlation between the severity of vessel disease and the severity of lung disease (Figure 2; p<0.001).’

Reviewer 2 Report

In this retrospective cohort study the authors explored pulmonary and lung features in children with PVS using contrast-enhanced CT and identified factors associated with death of these patients. The paper is well written, the methods are appropriate and the results are clear and of interest.

Minor comments:

Introduction is rather poor. I suggest the authors to enrich the text by adding data prevalence/incidence of PVS in the USA and at the global level, and the trend of incidence of this congenital defect over the years.  Furthermore, please note that the full name of CT given in the last sentence of Introduction was already provided above in the text.

Author Response

Reviewer #2:

Point 1:

Introduction is rather poor. I suggest the authors to enrich the text by adding data prevalence/incidence of PVS in the USA and at the global level, and the trend of incidence of this congenital defect over the years.

Response 1:

Thank you for making this suggestion. We agree that the introduction could have been written with more clarity about the prevalence/incidence of PVS in the USA and at the global level. We revised the introduction to include the following 2 sentences (page 1 Introduction, lines 3-5 and 6-8):

‘The incidence of PVS in children is reported to be in the range of 0.0017-0.03%, with a bimodal distribution of age at diagnosis, with 18 months as a dividing point.’

‘Although PVS has been well described in high income countries, recent data has shown that PVS also affects children in low-to-middle income countries, with a high mortality.’

Point 2:

Furthermore, please note that the full name of CT given in the last sentence of Introduction was already provided above in the text.

Response 2:

Thank you for identifying this duplicate annotation. We removed ‘computed tomography’ from the last sentence of the introduction (page 1, last line).